# Machine Teaching of Active Sequential Learners

**Tomi Peltola**
tomi.peltola@aalto.fi

**Mustafa Mert Çelikok**
mustafa.celikok@aalto.fi

**Pedram Daee**
pedram.daee@aalto.fi

**Samuel Kaski**
samuel.kaski@aalto.fi
Helsinki Institute for Information Technology HIIT
Department of Computer Science, Aalto University, Helsinki, Finland

## Abstract

Machine teaching addresses the problem of finding the best training data that can guide a learning algorithm to a target model with minimal effort. In conventional settings, a teacher provides data that are consistent with the true data distribution. However, for sequential learners which actively choose their queries, such as multi-armed bandits and active learners, the teacher can only provide responses to the learner's queries, not design the full data. In this setting, consistent teachers can be sub-optimal for finite horizons. We formulate this sequential teaching problem, which current techniques in machine teaching do not address, as a Markov decision process, with the dynamics nesting a model of the learner and the actions being the teacher's responses. Furthermore, we address the complementary problem of learning from a teacher that plans: to recognise the teaching intent of the responses, the learner is endowed with a model of the teacher. We test the formulation with multi-armed bandit learners in simulated experiments and a user study. The results show that learning is improved by (i) planning teaching and (ii) the learner having a model of the teacher. The approach gives tools to taking into account strategic (planning) behaviour of users of interactive intelligent systems, such as recommendation engines, by considering them as boundedly optimal teachers.

## 1 Introduction

Humans, casual users and domain experts alike, are increasingly interacting with artificial intelligence or machine learning based systems. As the number of interactions in human–computer and other types of agent–agent interaction is usually limited, these systems are often based on active sequential machine learning methods, such as multi-armed bandits, Bayesian optimization, or active learning. These methods explicitly optimise for the efficiency of the interaction from the system's perspective. On the other hand, for goal-oriented tasks, humans create mental models of the environment for planning their actions to achieve their goals [1, 2]. In AI systems, recent research has shown that users form mental models of the AI's state and behaviour [3, 4]. Yet, the statistical models underlying the active sequential machine learning methods treat the human actions as passive data, rather than acknowledging the strategic thinking of the user.

Machine teaching studies a complementary problem to active learning: how to provide a machine learner with data to learn a target model with minimal effort [5–7]. Apart from its fundamental machine learning interest, machine teaching has been applied to domains such as education [8] and adversarial attacks [9]. In this paper, we study the machine teaching problem of active sequential machine learners: the learner sequentially chooses queries and the teacher provides responses to them. Importantly, to steer the learner towards the teaching goal, the teacher needs to appreciate the order of the learner's queries and the effect of the responses on it. Current techniques in machine teaching

do not address such interaction. Furthermore, by viewing users as boundedly optimal teachers, and solving the (inverse machine teaching) problem of how to learn from the teacher's responses, our approach provides a way to formulate models of strategically planning users in interactive AI systems.

Our main contributions are (i) formulating the problem of machine teaching of active sequential learners as planning in a Markov decision process, (ii) formulating learning from the teacher's responses as probabilistic inverse reinforcement learning, (iii) implementing the approach in Bayesian Bernoulli multi-armed bandit learners with arm dependencies, and (iv) empirically studying the performance in simulated settings and a user study. Source code is available at `https://github.com/AaltoPML/machine-teaching-of-active-sequential-learners`.

## 2 Related work

Most work in machine teaching considers a batch setting, where the teacher designs a minimal dataset to make the learner learn the target model [5–7]. Some works have also studied sequential teaching, but in different settings from ours: Teaching methods have been developed to construct batches of state-action trajectories for inverse reinforcement learners [10, 11]. Variations on teaching online learners, such as gradient descent algorithms, by providing them with a sequence of $(x, y)$ data points have also been considered [12–14]. Teaching in the context of education, with uncertainty about the learner's state, has been formulated as planning in partially-observable Markov decision processes [8, 15]. A theoretical study of the teacher-aware learners was presented in [16, 17] where the teacher and the learner are aware of their cooperation. Compared to our setting, in these works, the teacher is in control of designing all of the learning data (while possibly using interaction to probe the state of the learner) and is not allowed to be inconsistent with regard to the true data distribution. Apart from [11, 16, 17], they also do not consider teacher-aware learners. Machine teaching can also be used towards attacking learning systems [9], and adversarial attacks against multi-armed bandits have been developed, by poisoning historical data [18] or modifying rewards online [19]. The goal, settings, and proposed methods differ from ours. Relatedly, our teaching approach for the case of a bandit learner can been seen as a form of reward shaping, which aims to make the environment more supportive of reinforcement learning by alleviating the temporal credit assignment problem [20].

The proposed model of the interaction between a teacher and an active sequential learner is a probabilistic multi-agent model. It can be connected to the overarching framework of interactive partially observable Markov decision processes (I-POMDPs; see Supplementary Section 1 for more details) [21] and other related multi-agent models [22–25]. I-POMDPs provide, in a principled decision-theoretic framework, a general approach to define multi-agent models that have recursive beliefs about other agents. This also forms a rich basis for computational models of theory of mind, which is the ability to attribute mental states, such as beliefs and desires, to oneself and other agents and is essential for efficient social collaboration [26, 27]. Our teaching problem nests a model of a teacher-unaware learner, forming a learner–teacher model. Teaching-aware learning adds a further layer, forming a nested learner–teacher–learner model, where the higher level learner models a teacher modelling a teaching-unaware learner. Learning from humans with recursive reasoning was opined in [28]. To our knowledge, our work is the first to propose a multi-agent recursive reasoning model in the practically important case of multi-armed bandits, allowing us to learn online from the scarce data emerging from human–computer interaction.

User modelling in human–computer interaction aims at improving the usability and usefulness of collaborative human–computer systems and providing personalised user experiences [29]. Machine learning based interactive systems extend user modelling to encompass statistical models interpreting user's actions. For example, in information exploration and discovery, the system needs to iteratively recommend items to the user and update the recommendations based on the user feedback [30, 31]. The current underlying statistical models use the user's response to the system's queries, such as *did you like this movie?*, as data for building a relevance profile of the user. Recent works have investigated more advanced user models [32, 33]; however, as far as we know, no previous work has proposed statistical user models that incorporate a model of the user's mental model of the system.

Finally, our approach can be grounded to computational rationality, which models human behaviour and decision making under uncertainty as expected utility maximisation, subject to computational constraints [34]. Our model assumes that the teacher chooses actions proportional to their likelihood to maximise, for a limited horizon, the future accumulated utility.

# 3 Model and computation

We consider machine teaching of an active sequential learner, with the iterations consisting of the learner querying an input point $x$ and the teacher providing a response $y$. First, the teaching problem is formulated as a Markov decision process, the solution of which provides a teaching policy. Then, learning from the responses provided by the teacher is formulated as an inverse reinforcement learning problem. We formulate the approach for general learners, and give a detailed implementation for the specific case of a Bayesian Bernoulli multi-armed bandit learner, which models arm dependencies.

## 3.1 Active sequential learning

Before considering machine teaching, we first define the type of active sequential learners considered. This also provides a baseline to which the teacher's performance is compared. The general definition encompasses multiple popular sequential learning approaches, including Bayesian optimisation and multi-armed bandits, which aim to learn fast, with few queries.

An active sequential learner is defined by (i) a machine learning model relating the response $y$ to the inputs $x$ through a function $f$, $y = f_{\theta}(x)$, parameterised by $\theta$, or through a conditional distribution $p(y \mid x, \theta)$, (ii) a deterministic learning algorithm, fitting the parameters $\theta$ or their posterior $p(\theta \mid \mathcal{D})$ given a dataset $\mathcal{D} = \{(x_1, y_1), \ldots, (x_t, y_t)\}$, (iii) a query function that, possibly stochastically, chooses an input point $x$ to query for a response $y$, usually formulated as utility maximisation.

The dynamics of the learning process then, for $t = 1, \ldots, T$, consists of iterating the following steps:

1. Use the query function to choose a query $x_t$.
2. Obtain the response $y_t$ for the query $x_t$ from a teacher (or some other information source).
3. Update the training set to $\mathcal{D}_t = \mathcal{D}_{t-1} \cup \{(x_t, y_t)\}$ and the model correspondingly.

The data produced by the dynamics forms a sequence, or *history*, $h_T = x_1, y_1, x_2, y_2, \ldots, x_T$ (we define the history to end at the input $x_T$, before $y_T$, for notational convenience in the following).

**Bayesian Bernoulli multi-armed bandit learner**   As our main application in this paper, we consider Bayesian Bernoulli bandits. At each iteration $t$, the learner chooses an arm $i_t \in \{1, \ldots, K\}$ and receives a stochastic reward $y_t \in \{0, 1\}$, depending on the chosen arm. The goal of the learner is to maximise the expected accumulated reward $R_T = \mathrm{E}[\sum_{t=1}^{T} y_t]$. This presents an exploration–exploitation problem, as the learner needs to learn which arms produce reward with high probability.

The learner associates each arm $k$ with a feature vector $x_k \in \mathbb{R}^M$ and models the rewards as Bernoulli-distributed binary random variables

$$p_{\mathcal{B}}(y_t \mid \mu_{i_t}) = \mathrm{Bernoulli}(y_t \mid \mu_{i_t}) \tag{1}$$

with reward probabilities $\mu_k = \sigma(x_k^{\mathrm{T}}\theta), k = 1, \ldots, K$, where $\theta \in \mathbb{R}^M$ is a weight vector and $\sigma(\cdot)$ the logistic sigmoid function. The linearity assumption could be relaxed, for example, by encoding the $x_k$'s using suitable basis functions or Gaussian processes. The Bayesian learner has a prior distribution on the model parameters, here assumed to be a multivariate normal, $\theta \sim \mathrm{N}(\mathbf{0}, \tau^2 \mathbf{I})$, with mean zero and diagonal covariance matrix $\tau^2 \mathbf{I}$. Given a collected set of arm selections and reward observations at step $t$, $\mathcal{D}_t = \{(i_1, y_1), \ldots, (i_t, y_t)\}$ (or equivalently $\mathcal{D}_t = (h_t, y_t)$), the posterior distribution of $\theta$, $p(\theta \mid \mathcal{D}_t)$ is computed.

The learner uses a bandit arm selection strategy to select the next arm to query about. Here, we use Thompson sampling [35], a practical and empirically and theoretically well-performing algorithm [36]; other methods could easily be used instead. The next arm is sampled with probabilities proportional to the arm maximising the expected reward, estimated over the current posterior distribution:

$$\Pr(i_{t+1} = k) = \int I(\arg\max_j \mu_j = k \mid \theta) p(\theta \mid \mathcal{D}_t) d\theta, \tag{2}$$

where $I$ is the indicator function. This can be realised by first sampling a weight vector $\theta$ from $p(\theta \mid \mathcal{D}_t)$, computing the corresponding $\mu^{(\theta)}$, and choosing the arm with the maximal reward probability, $i_{t+1} = \arg\max_k \mu_k^{(\theta)}$.

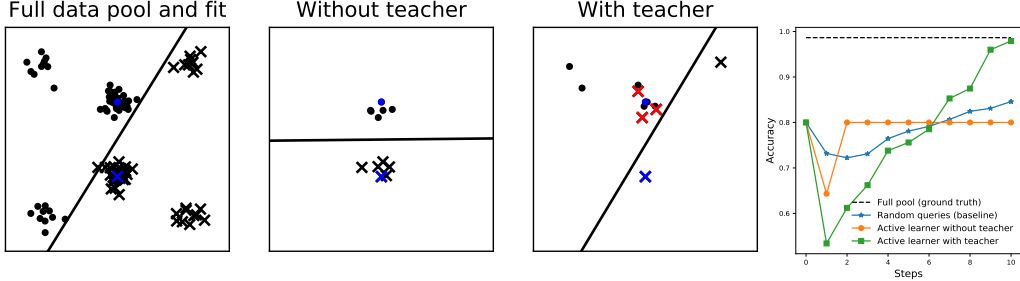

Figure 1: Example of teaching effect on pool-based logistic regression active learner. Using uncertainty sampling for queries, the learner fails to sample useful points from the pool in 10 iterations to learn a good decision boundary ("Without teacher"; starting from blue training data). A planning teacher can help the learner sample more representative points by switching some labels ("With teacher"; switched labels are shown in red). The average accuracy improvement is shown in the right panel. Details of the setting are given in Supplementary Section 2.

## 3.2 Machine teaching of active sequential learner

In standard active sequential learning, the responses $y_t$ are assumed to be generated by a stationary data-generating mechanism as independent and identically distributed samples. We call such a mechanism a *naive teacher*. Our machine teaching formulation replaces it with a *planning teacher* which, by choosing $y_t$ carefully, aims to steer the learner towards a teaching goal with minimal effort.

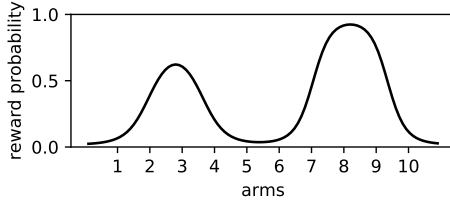

We formulate the teaching problem as a Markov decision process (MDP), where the transition dynamics follow from the dynamics of the sequential learner and the responses $y_t$ are the actions. The teaching MDP is defined by the tuple $\mathcal{M} = (\mathcal{H}, \mathcal{Y}, \mathcal{T}, \mathcal{R}, \gamma)$, where states $h_t \in \mathcal{H}$ correspond to the history, actions are the responses $y_t \in \mathcal{Y}$, transition probabilities $p(h_{t+1} \mid h_t, y_t) \in \mathcal{T}$ are defined by the learner's sequential dynamics, rewards $R_t(h_t) \in \mathcal{R}$ are used to define the teacher's goal, and $\gamma \in (0, 1]$ is a dis-

Figure 2: Example of the teaching effect on a multi-armed bandit learner. With the environmental reward probabilities shown in the figure, consider the first query being arm 6. The reward probability for the arm is low, so $y_1 = 0$ with high probability for a naive teacher. Yet, the optimal action for a planning teacher is $y_1 = 1$, because the teacher can anticipate that this will lead to a higher probability for the learner to sample the next arm near the higher peak. Details on the setting are given in Supplementary Section 3.

count factor (optional if $T$ is finite). The objective of the teacher is to choose actions $y_t$ to maximise the cumulative reward, called value, $V^\pi(h_1) = \mathrm{E}^\pi[\sum_{t=1}^{T} \gamma^{t-1} R_t(h_t)]$, where $T$ is the teacher's planning horizon and the expectation is over the possible stochasticity in the learner's queries and the teacher's policy. The teacher's policy $\pi(h_t, y_t) = p(y_t \mid h_t, \pi)$ maps the state $h_t$ to probabilities over the action space $\mathcal{Y}$. The solution to the teaching problem corresponds to finding the optimal teaching policy $\pi^*$.

The reward function $R_t(h_t)$ defines the goal of the teacher. In designing a teaching MDP, as in reinforcement learning, its choice is crucial. In machine teaching, a natural assumption is that the reward function is parameterized by an optimal model parameter $\boldsymbol{\theta}^*$, or some other ground truth, known to the teacher but not the learner. For teaching of a supervised learning algorithm, the reward $R_t(h_t; \boldsymbol{\theta}^*)$ can, for example, be defined based on the distance of the learner's estimate of $\boldsymbol{\theta}$ to $\boldsymbol{\theta}^*$ or by evaluation of learner's predictions against the teacher's privileged knowledge of outcomes (Figure 1).

In the multi-armed bandit application, it is assumed that the teacher knows the true parameter $\boldsymbol{\theta}^*$ of the underlying environmental reward distribution and aims to teach the learner such that the accumulated environmental reward is maximised (Figure 2). We define the teacher's reward function as $R_t(h_t; \boldsymbol{\theta}^*) = \boldsymbol{x}_t^\mathrm{T} \boldsymbol{\theta}^*$ (leaving out $\sigma(\cdot)$ to simplify the formulas for the teacher model).

**Properties of the teaching MDP** In Supplementary Section 4, we briefly discuss the transition dynamics and state definition of the teaching MDP, and contrast it to Bayes-adaptive MDPs to better understand its properties. Finding the optimal teaching policy presents similar challenges to planning in Bayes-adaptive MDPs. Methods such as Monte Carlo tree search [37] have been found to provide effective approaches.

### 3.3 Learning from teacher's responses

We next describe how the learner can interpret the teacher's responses, acknowledging the teaching intent. Having formulated the teaching as an MDP, the teacher-aware learning follows naturally as inverse reinforcement learning [38, 39]. We formulate a probabilistic teacher model to make the learning more robust towards suboptimal teaching and to allow using the teacher model as a block in probabilistic modelling.

At each iteration $t$, the learner assumes that the teacher chooses the action $y_t$ with probability proportional to the action being optimal in value:

$$p_{\mathcal{M}}(y_t \mid h_t, \boldsymbol{\theta}^*) = \frac{\exp\left(\beta Q^*(h_t, y_t; \boldsymbol{\theta}^*)\right)}{\sum_{y' \in \mathcal{Y}} \exp\left(\beta Q^*(h_t, y'; \boldsymbol{\theta}^*)\right)}, \tag{3}$$

where $Q^*(h_t, y_t; \boldsymbol{\theta}^*)$ is the optimal state-action value function of the teaching MDP for the action $y_t$ (that is, the value of taking action $y_t$ at $t$ and following an optimal policy afterwards). Here $\beta$ is a *teacher optimality parameter* (or inverse temperature; for $\beta = 0$, the distribution of $y_t$ is uniform; for $\beta \to \infty$, the action with the highest value is chosen deterministically). From the teaching-aware learner's perspective, the teacher's $\boldsymbol{\theta}^*$ is unknown, and Equation 3 functions as the likelihood for learning about $\boldsymbol{\theta}$ from the observed teaching. In the bandit case, this replaces Equation 1. Note that the teaching MDP dynamics still follow from the teaching-unaware learner.

**One-step planning** Since our main motivating application is modelling users as boundedly optimal teachers, implemented for a Bernoulli multi-armed bandit system, it is interesting to consider the special case of one-step planning horizon, $T = 1$. The state-action value function $Q^*(h_t, y_t; \boldsymbol{\theta}^*)$ then simplifies to the rewards at the next possible arms, and the action observation model to

$$p_{\mathcal{M}}(y_t \mid h_t, \boldsymbol{\theta}^*) \propto \exp(\beta((\boldsymbol{\theta}^*)^{\mathrm{T}} \boldsymbol{X}^{\mathrm{T}} \boldsymbol{p}_{h_t, y_t})), \tag{4}$$

where $\boldsymbol{p}_{h_t, y_t} = [p_{1, h_t, y_t}, \ldots, p_{K, h_t, y_t}]^{\mathrm{T}}$ collects the probabilities of the next arm given action $y_t \in \{0, 1\}$ at the current arm $\boldsymbol{x}_t$ in $h_t$, as estimated according to the teaching MDP, and $\boldsymbol{X} \in \mathbb{R}^{K \times M}$ collects the arm features into a matrix. Note that the reward of the current arm does not appear in the action probability[1]. For deterministic bandit arm selection strategies, the transition probabilities $p_{k, h_t, y_t}$ for each of the two actions would have a single 1 and $K - 1$ zeroes (essentially picking one of the possible arms), giving the action probability an interpretation as a preference for one of the possible next arms. For stochastic selection strategies, such as Thompson sampling, the interpretation is similar, but the two arms are now weighted averages, $\bar{\boldsymbol{x}}_{y_t=0} = \boldsymbol{X}^{\mathrm{T}} \boldsymbol{p}_{h_t, y_t=0}$ and $\bar{\boldsymbol{x}}_{y_t=1} = \boldsymbol{X}^{\mathrm{T}} \boldsymbol{p}_{h_t, y_t=1}$. An algorithmic overview of learning with a one-step planning teacher model is given in Supplementary Section 5.

For an illustrative example, consider a case with two independent arms ($\boldsymbol{x}_1 = [1, 0]$ and $\boldsymbol{x}_2 = [0, 1]$), with the first arm having a larger reward probability than the other ($\theta_1^* > \theta_2^*$). The optimal teaching action is then to give $y_t = 1$ for queries on arm 1 and $y_t = 0$ for arm 2. A teaching-unaware learner will still need to query both arms multiple times to identify the better arm. A teaching-aware learner (when $\beta \to \infty$) can identify the better arm from a single query (on either arm), since the likelihood function tends to the step function $I(\theta_1^* > \theta_2^*)$. This demonstrates that the teaching-aware learner can use a query to reduce uncertainty about other arms even in the extreme case of independent arms.

**Incorporating uncertainty about the teacher** Teachers can exhibit different kinds of strategies. To make the learner's model of the teacher robust to different types of teachers, we formulate a mixture model over a set of alternative strategies. Here, for the multi-armed bandit case, we consider a combination of a teacher that just passes on the environmental reward (naive teacher, Equation 1) and the planning teacher (Equation 3):

$$p_{\mathcal{B}/\mathcal{M}}(y_t \mid h_t, \boldsymbol{\theta}^*, \alpha) = (1 - \alpha) p_{\mathcal{B}}(y_t \mid \mu_{i_t}) + \alpha p_{\mathcal{M}}(y_t \mid h_t, \boldsymbol{\theta}^*), \tag{5}$$

where $\alpha \in (0, 1)$ is a mixing weight and $\mu_{i_t} = \sigma(\boldsymbol{x}_{i_t}^{\mathrm{T}} \boldsymbol{\theta}^*)$ is the reward probability of the latest arm in the history $h_t$. A beta prior distribution, $\alpha \sim \text{Beta}(1, 1)$, is assumed for the mixing weight.

### 3.4 Computational details for Bayesian Bernoulli multi-armed bandits

Computation presents three challenges: (i) computing the analytically intractable posterior distribution of the model parameters $p(\boldsymbol{\theta} \mid \mathcal{D}_t)$ or $p(\boldsymbol{\theta}^*, \alpha \mid \mathcal{D}_t)$, (ii) solving the state-value functions $Q^*$ for the teaching MDP, and (iii) computing the Thompson sampling probabilities that are needed for the state-value functions.

We implemented the models in the probabilistic programming language Pyro (version 0.3, under PyTorch v1.0) [40] and approximate the posterior distributions with Laplace approximations [41, Section 4.1]. In brief, the posterior is approximated as a multivariate Gaussian, with the mean defined by the maximum a posteriori (MAP) estimate and the covariance matrix being the negative of the inverse Hessian matrix at the MAP estimate. In the mixture model, the mixture coefficient $\alpha \in (0, 1)$ is transformed to the real axis via the logit function before computing the approximation.

The inference requires computing the gradient of the logarithm of the unnormalised posterior probability. For the teacher model, this entails computing the gradient of the logarithm of Equation 3 at any value of the model parameters, which requires solving and computing the gradients of the optimal state-action value functions $Q^*$ with respect to $\boldsymbol{\theta}^*$. To solve the $Q^*$ for both of the possible observable actions $y_t = 0$ and $y_t = 1$, we compute all the possible trajectories in the MDP until the horizon $T$ and choose the ones giving maximal expected cumulative reward. Choi and Kim [39] show that the gradients of $Q^*$ exist almost everywhere, and that the direct computation gives a subgradient at the boundaries where the gradient does not exist.

We mainly focus on one-step planning ($T = 1$) in the experiments. For long planning horizons and stochastic arm selection strategies, the number of possible trajectories grows too fast for the exact exhaustive computation to be feasible ($K^T$ trajectories for each initial action). In our multi-step experiments, we approximate the forward simulation of the MDP with *virtual arms*: instead of considering all possible next arms given an action $y_t$ and weighting them with their selection probabilities $\boldsymbol{p}_{h_t, y_t}$, we update the model with a virtual arm that is the selection-probability-weighted average of the next possible arms $\bar{\boldsymbol{x}}_{h_t, y_t} = \boldsymbol{X}^{\mathrm{T}} \boldsymbol{p}_{h_t, y_t}$ (for deterministic strategies, this is exact computation). The virtual arms do not correspond to real arms in the system but are expectations over the next arms. This leads to $2^{T-1}$ trajectories to simulate for each initial action. Moreover, for any trajectory of actions $y_1, \ldots, y_T$, this approximation gives $Q(h_1, y_1; \boldsymbol{\theta}^*) \approx (\boldsymbol{\theta}^*)^{\mathrm{T}} \boldsymbol{X}^{\mathrm{T}} \sum_{t=1}^{T} \gamma^{t-1} \boldsymbol{p}_{h_t, y_t}$ and if we cache the sum of the discounted transition probabilities for each trajectory from the forward simulation, we can easily find the optimal $Q^*$ at any value of $\boldsymbol{\theta}^*$ as required for the inference.

Computing the next arm probabilities for the $Q^*$ values requires computing the actual Thompson sampling probabilities in Equation 2 instead of just sampling from it. As the sigmoid function is monotonic, one can equivalently compute the probabilities as $\Pr(i_{t+1} = k) = \int I(\arg\max_j z_j = k) p(\boldsymbol{z} \mid \mathcal{D}_t) d\boldsymbol{z}$ where $\boldsymbol{z} = \boldsymbol{X} \boldsymbol{\theta}^*$. As $p(\boldsymbol{\theta}^* \mid \mathcal{D}_t) \approx \mathrm{N}(\boldsymbol{\theta}^* \mid \boldsymbol{m}, \boldsymbol{\Sigma})$, $\boldsymbol{z}$ has multivariate normal distribution with mean $\boldsymbol{X} \boldsymbol{m}$ and covariance $\boldsymbol{X} \boldsymbol{\Sigma} \boldsymbol{X}^{\mathrm{T}}$. The selection probabilities can then be estimated with Monte Carlo sampling. We further use Rao-Blackwellized estimates $\Pr(i_{t+1} = k) \approx \frac{1}{L} \sum_{l=1}^{L} \Pr(z_k > \max_{j \neq k} z_j \mid \boldsymbol{z}_{-k}^{(l)})$, with $L$ Monte Carlo samples drawn for $\boldsymbol{z}_{-k}$ ($\boldsymbol{z}$ with $k$th component removed) and $\Pr(z_k > \max_{j \neq k} z_j \mid \boldsymbol{z}_{-k}^{(l)})$ being the conditional normal probability of component $z_k$ being larger than the largest component in $\boldsymbol{z}_{-k}$.

## 4 Experiments

We perform simulation experiments for the Bayesian Bernoulli multi-armed bandit learner, based on a real dataset, to study (i) whether a teacher can efficiently steer the learner towards a target to increase learning performance, (ii) whether the ability of the learner to recognise the teaching intent increases the performance, (iii) whether the mixture model is robust to assumptions about the teacher's strategy, and (iv) whether planning multiple steps ahead improves teaching performance. We then present results from a proof-of-concept study with humans. Supplementary Section 6.1 includes an additional experiment studying the teaching of an uncertainty-sampling-based logistic regression active learner, showing that teaching can improve learning performance markedly.

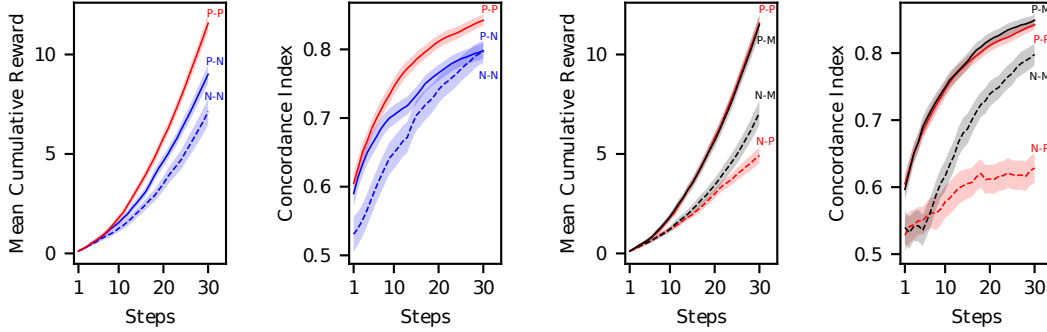

Figure 3: **Left-side panels:** Planning teacher improves performance, both when the learner's teacher model is naive (P-N) or planning (P-P), over naive teacher (N-N). **Right-side panels:** Naive teacher with a learner expecting a planning teacher (N-P) degrades performance. Learners with the mixture teacher model attain similar performance to matched models (P-M vs P-P and N-M vs N-N (left)). Lines show the mean over 100 replications and shaded area the 95% confidence intervals for the mean. See Table 1 for key to the abbreviations.

## 4.1 Simulation experiments

We use a word relevance dataset for simulating an information retrieval task. In this task, the user is trying to teach a relevance profile to the learner in order to reach her target word. The Word dataset is a random selection of 10,000 words from Google's Word2Vec vectors, pre-trained on Google News dataset [42]. We reduce the dimensionality of the word embeddings from the original 300 to 10 using PCA. Feature vectors are mean-centred and normalised to unit length. We report results, with similar conclusions, on two other datasets in Supplementary Section 6.2.

We randomly generate 100 replicate experiments: a set of 100 arms is sampled without replacement and one arm is randomly chosen as the target $\hat{\boldsymbol{x}} \in \mathbb{R}^M$. The ground-truth relevance profile is generated by first setting $\hat{\boldsymbol{\theta}}^* = [c, d\hat{\boldsymbol{x}}] \in \mathbb{R}^{M+1}$, where $c = -4$ is a weight for an intercept term (a constant element of 1 is added to the $\boldsymbol{x}$s) and $d = 8$ is a scaling factor. Then, the ground-truth reward probabilities are computed as $\hat{\mu}_k = \sigma(\boldsymbol{x}_k^T \hat{\boldsymbol{\theta}}^*)$ for each arm $k$ (Supplementary Figure 2 shows the mean reward probability profile). To reduce experimental variance for method comparison, we choose one of the arms randomly as the initial query for all methods.

We compare the learning performances of different pairs of simulated teachers and learners (Table 1). A naive teacher (N), which does not intentionally teach, passes on a stochastic binary reward (Equation 1) based on the ground truth $\hat{\mu}_k$ as its action for arm $k$ (the standard bandit assumption). A planning teacher (P) uses the probabilistic teaching MDP model (Equation 4

Table 1: Teacher–learner pairs.

| **Teacher** | **Learner's model of teacher** | | |
|---|---|---|---|
| | naive | planning | mixture |
| naive | N-N | N-P | N-M |
| planning | P-N | P-P | P-M |

for one-step and Equation 3 for multi-step) based on the ground truth $\hat{\boldsymbol{\theta}}^*$ to plan its action. We use $\hat{\beta} = 20$ as the planning teacher's optimality parameter and also set $\beta$ of the learner's teacher model to the same value. For multi-step models, we set $\gamma_t = \frac{1}{T}$, so that they plan to maximise the average return up to horizon $T$. The learners are named based on their models of the teacher: a teaching-unaware learner learns based on the naive teacher model (N; Equation 1) and teaching-aware learner models the planning teacher (P; Equation 4 or Equation 3). Mixture model (M) refers to the learner with a mixture of the two teacher models (Equation 5).

Expected cumulative reward and concordance index are used as performance measures (higher is better for both). Expected cumulative reward measures how efficiently the system can find high reward arms and is a standard bandit benchmark value. Concordance index is equivalent to the area under the receiver operating characteristic curve. It is a common performance measure for information retrieval tasks. It estimates the probability that a random pair of arms is ordered in the same order by their ground truth relevances and the model's estimated relevances; 0.5 corresponds to random and 1.0 to perfect performance.

## 4.2 Simulation results

**Teaching improves performance** Figure 3 shows the performance of different combinations of pairs of teachers and learners (where planning teachers have planning horizon $T = 1$). The planning teacher can steer a teacher-unaware learner to achieve a marked increase in performance compared to a naive teacher (P-N vs N-N; left-side panels), showing that intentional teaching makes the reward signal more supportive of learning. The performance increases markedly further when the learner models the planning teacher (P-P; left-side panels). The improvements are seen in both performance measures, and the concordance index implies particularly that the proposed model learns faster about relevant arms and also achieves higher overall performance at the end of the 30 steps.

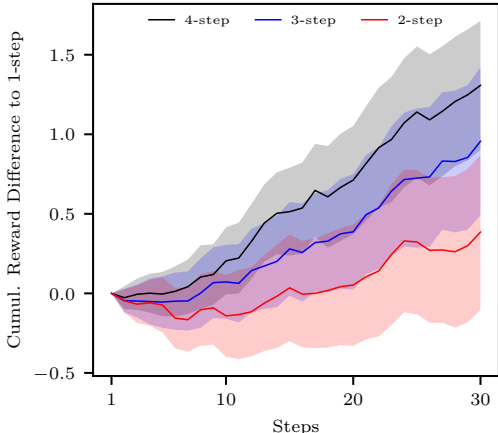

Figure 4: Teachers planning for multiple steps ahead improve over 1-step (P-P) in performance.

**Mixture model increases robustness to assumptions about the teacher** A mismatch of a naive teacher with a learner expecting a planning teacher (N-P) is markedly detrimental to performance (Figure 3 right-side panels). The mixture model guards against the mismatch and attains a performance similar to the matching assumptions (P-M vs P-P and N-M vs N-N).

**Planning for multiple steps increases performance further** Figure 4 shows the cumulative reward difference for matching planning teacher–learner pairs (P-P) when planning two to four steps ahead compared to one step. There is a marked improvement especially when going to 3-step or 4-step planning horizon.

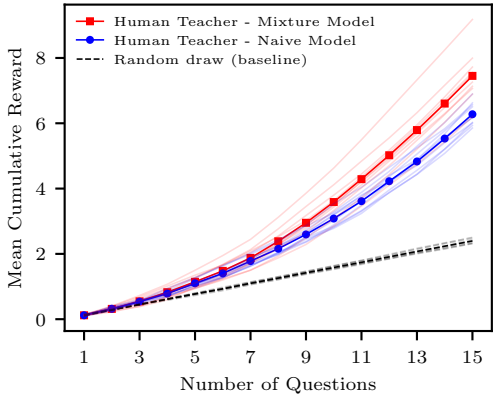

Figure 5: The accumulated reward was consistently higher for the participants when interacting with a learner having the mixture teacher model, compared to a learner with the naive teacher model. Shaded lines show the mean performance (over the 20 target words) of individual participants. Solid lines show the mean over the participants. Random arm sampling is shown as baseline.

**Sensitivity analysis** Sensitivity of the results to the simulated teacher's optimality parameter $\hat{\beta}$ (performance degrades markedly for small values of $\hat{\beta}$) and to the number of arms (500 instead of 100; results remain qualitatively similar) are shown in Supplementary Section 6.2.

## 4.3 User experiment

We conducted a proof-of-concept user study for the task introduced above, using a subset of 20 words on ten university students and researchers. The goal of the study was introduced to the participants as helping a system to find a target word, as fast as possible, by providing binary answers (yes/no) to the system's questions: "Is this word relevant to the target?" A target word was given to the participants at the beginning of each round (for twenty rounds; each word chosen once as the target word). Details of the study setting are provided in Supplementary Section 7.

Participants achieved noticeably higher average cumulative reward when interacting with a learner having the mixture teacher model, compared to a learner with the naive teacher model (Figure 5, red vs blue). This difference was at a significant level (p-value < 0.01) after 12 questions, computed using paired sample t-test (see Supplementary Section 7 for p-values per step).

# 5 Discussion and conclusions

We introduced a new sequential machine teaching problem, where the learner actively chooses queries and the teacher provides responses to them. This encompasses teaching popular sequential learners, such as active learners and multi-armed bandits. The teaching problem was formulated as a Markov decision process, the solution of which provides the optimal teaching policy. We then formulate teacher-aware learning from the teacher's responses as probabilistic inverse reinforcement learning. Experiments on Bayesian Bernoulli multi-armed bandits and logistic regression active learners demonstrated improved performance from teaching and from learning with teacher awareness. Better theoretical understanding of the setting and studying a more varied set of assumptions and approaches to planning for both the teacher and the teacher-aware learner are important future directions.

Our formulation provides a way to model users with strategic behaviour as boundedly optimal teachers in interactive intelligent systems. We conducted a proof-of-concept user study, showing encouraging results, where the user was tasked to steer a bandit system towards a target word. To scale the approach to more realistic systems, for example, to interactive exploratory information retrieval [43], of which our user study is a simplified instance, or to human-in-the-loop Bayesian optimisation [44], where the user might not possess the exact knowledge of the goal, future work should consider incorporating more advanced cognitive models of users. As an efficient teacher (user) needs to be able to model the learner (system), our results also highlight the role of understandability and predictability of interactive systems for the user as an important design factor, not only for user experience, but also for the statistical modelling in the system.

While we focused here on teachers with bounded, short-horizon planning (as we would not expect human users to be able to predict behaviour of interactive systems for long horizons), scaling the computation to larger problems is of interest. Given the similarity of the teaching MDP to Bayes-adaptive MDPs (and partially observable MDPs), planning methods developed for them could be used for efficient search for teaching actions. The teaching setting has some advantages here: as the teacher is assumed to have privileged information, such as a target model, that information could be used to generate a reasonable initial policy for choosing actions $y$. Such policy could be then refined, for example, using Monte Carlo tree search. The teacher-aware learning problem is more challenging, as inverse reinforcement learning requires handling the planning problem in an inner loop. Considering the application and adaptation of state-of-the-art inverse reinforcement learning methods for teacher-aware learning is future work.

### Acknowledgments

This work was financially supported by the Academy of Finland (Flagship programme: Finnish Center for Artificial Intelligence, FCAI; grants 319264, 313195, 305780, 292334). Mustafa Mert Çelikok is partially funded by the Finnish Science Foundation for Technology and Economics KAUTE. We acknowledge the computational resources provided by the Aalto Science-IT Project. We thank Antti Oulasvirta and Marta Soare for comments that improved the article.

## Footnotes

[1] It cancels out. The teacher cannot affect the arm choice anymore, as it has already been made.

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
