[Supplementary Material]

# *Supplementary Materials to*
# Machine Teaching of Active Sequential Learners

**Tomi Peltola**
tomi.peltola@aalto.fi

**Mustafa Mert Çelikok**
mustafa.celikok@aalto.fi

**Pedram Daee**
pedram.daee@aalto.fi

**Samuel Kaski**
samuel.kaski@aalto.fi
Helsinki Institute for Information Technology HIIT
Department of Computer Science, Aalto University, Helsinki, Finland

## 1 Connection to I-POMDPs and multi-agent opponent modelling

Recursive modelling of the opponents' reasoning is studied in multi-agent and game theory communities. These methods employ theory-of-mind-like models to reason about the opponents' behaviour.

Interactive POMDPs are a general recursive model of multi-agent interaction where the state space of a POMDP is extended by adding possible models of the opponents [1]. An I-POMDP agent maintains a belief over the original states of the POMDP and the possible models of its opponents. Opponent models are nested in the sense that a level-$k$ I-POMDP has level-$k-1$ opponent models in its state space. Level $0$ is a POMDP with no opponent models, where the effects of the actions of others are subsumed into transition dynamics.

I-POMDPs suffer from three curses: (1) curse of dimensionality; because the state space is over the states and joint beliefs, (2) curse of history; because the policy space grows exponentially with respect to the planning horizon, and (3) curse of nestedness. The curse of nestedness is due to the fact that the solution of each level depends recursively on the solutions of lower-level I-POMDPs. Even though there are approximate particle filtering approaches, previous work has shown that they scale poorly even to medium-sized problems [2]. So far, most of the recent work in I-POMDPs has been evaluated only on toy domains [3].

Our proposed learner with the planning teacher model is a level-1 agent who models its teacher as a level-0 agent. The level-0 agent is a teaching MDP who subsumes the query selection behaviour of the learner into the transition dynamics $\mathcal{T}^+$. Different from general I-POMDPs, we do not have any environment states. The teacher and learner are interacting directly. The level-0 agent models the learner as the environment, and the level-1 learner models the teacher as a level-0 agent. The learner's goal is to learn the function behind the teacher's actions (represented as the reward function) and its teacher model changes the learning rule via the likelihood.

The opponent model space of I-POMDPs usually contains an additional type of models called sub-intentional models. These are simple models such as an opponent who acts uniformly at random, or one that chooses its actions from a fixed yet unknown distribution. Our mixture model can be seen as a level-1 learner which maintains a belief over two possible opponent models: level-0 planning teacher model and the sub-intentional naive teacher model. This belief is represented by the posterior of the mixture coefficient $\alpha$.

In summary, our modelling can be seen as part of the I-POMDP framework, yet we differ in terms of objectives, modes of interaction, and environment settings. These differences allow us to have reasonable improvements in terms of computational and sample complexity.

## 2   Details for the example of teaching effect on pool-based logistic regression active learner

Figure 1 in the main text shows an example of teaching effect on a pool-based logistic regression active learner. The learner is a logistic regression model with L2 regularization. It is initialised with 2 data points (one for each label from the large clusters in the middle) and has a pool of 60 unlabeled data points for which it can query the label. The generated dataset follows a pattern where uncertainty sampling is known to fail [4]. Ten iterations are run in the example.

The learner uses uncertainty sampling for selecting queries: the next query is chosen as the data point $x$, for the label of which the current logistic regression model has the largest entropy, $-\sum_{y\in\{0,1\}} p_{\boldsymbol{\theta}}(y \mid x) \log p_{\boldsymbol{\theta}}(y \mid x)$. After obtaining a label, the model is updated and a new query made. Each unlabeled data point can be queried only once.

The teacher plans for the full horizon of 10 iterations, with the reward defined at the terminal state of the horizon as the accuracy of the classification. The teacher has knowledge of the labels for the full pool of data, so the accuracy for the reward is evaluated using the full pool.

## 3   Details for the example of the teaching effect on a multi-armed bandit learner

Figure 2 in the main text shows an example of the teaching effect on a multi-armed bandit learner. The example follows the Bayesian Bernoulli multi-armed bandit setting. The ten arms are located evenly spaced on the x-axis from $0$ to $1$ (in the main text, the figure's x-axis labels show the arm numbers). A three-dimensional feature space is constructed with the first feature being constant $1$, second based on an RBF kernel at $0.2$ on the x-axis, and the third an RBF kernel at $0.8$ on the x-axis. The reward function is linear in this feature space, with weight $\boldsymbol{\theta}^* = [-4, 4.5, 6.5]$.

Consider the first query, before the learner has any observations, being arm 6. The reward probability for the arm 6 is low (0.06), so $y_1 = 0$ with high probability when there is no teacher. Yet, the optimal action for a one-step planning teacher (i.e., teacher's horizon $T = 1$) is $y_1 = 1$, because the teacher can anticipate this leading to a higher probability of sampling the next arm near the higher peak.

## 4   Properties of the teaching MDP and comparison to Bayes-adaptive Markov decision processes

We briefly discuss the transition dynamics and state definition of the teaching MDP, and contrast it to Bayes-adaptive MDPs (BAMDPs) to better understand its properties. The teaching MDP is defined in Section 3.2 of the main text.

The transition probabilities $p(h_{t+1} \mid h_t, y_t)$ can be decomposed into two factors: (1) an update of the learner's model given the new observation $y_t$ at the query point $x_t$ (corresponding to step 3 in the learning dynamics), which contributes a deterministic factor to the transition (equivalent to adding $y_t$ to the history), and (2) the selection of a new query point $x_{t+1}$ using the query function (corresponding to step 1 in the learning dynamics). If the query function is stochastic, this contributes a stochastic factor. Otherwise, the transitions are deterministic.

Depending on the learner model, instead of defining the state as the full history of the process, it is possible to define the state as a combination of the latest query point $x_t$ and the model parameters (or posterior parameters), if the model parameters form a sufficient statistic for the history with respect to the model and the query function.

The structure of the teaching MDP is similar in two respects to Bayes-adaptive MDPs (and partially observable MDPs), which describe an agent's uncertainty about the underlying transition dynamics (or state) [5, 6]: (1) the state definition includes the full history of the process (or its sufficient statistic), and (2) the transition probabilities can be decomposed into the two steps, featuring an update of the model and sampling of a transition conditional on the model. The main difference is that the teaching MDP does not describe the agent's uncertainty about the process, but the dynamics themselves evolve according to the two steps. In particular, the teacher is assumed to know the next arm probabilities (so there is no uncertainty about the dynamics, although the dynamics can

be stochastic, so the teacher does not know the exact next arm). Moreover, the model updated in BAMDP is directly a model of the transition dynamics (with observations being of form $(s, a, s')$, that is, new state $s'$ given previous state $s$ and action $a$), whereas in the teaching MDP, it is the learner's model (with observations of form $(\boldsymbol{x}, y)$, response $y$ given a query point $\boldsymbol{x}$). The next arm probabilities then follow from the definition of the query algorithm, and, for Bayesian bandit learner, can be computed as expectations over the learner's posterior distribution. The teaching MDP could be naturally extended to a teaching BAMDP, if the teacher has uncertainty about the learner (for example, which query strategy the learner uses). In any case, similar challenges to BAMDPs are faced in finding the optimal policy. Planning methods, such as Monte Carlo tree search [7], have been found to provide effective approaches.

### 4.1 Further BAMDP background and using them to model multi-armed bandit problems

A Bayes-adaptive Markov decision process (BAMDP) extends the state space of an MDP with unknown transition dynamics $\mathcal{T}$ by adding posterior beliefs about the transition dynamics [5]. This new state is also called the information state. The information state of a BAMDP at a given time is $s^+ = (s, h)$ where $s$ is the original MDP state and $h$ is the history of transitions observed so far. Then, a belief over transitions $P(\mathcal{T} \mid h)$ is maintained. Often in practice, this is a parametric distribution and the information state maintains the sufficient statistics instead of the whole history. The transition dynamics of the BAMDP in the extended state space is $\mathcal{T}^+((s, h), a, (s', h')) = \mathbb{E}_{P(\mathcal{T}|h)}[\mathcal{T}(s, a, s')]$. BAMDP's actions and rewards are the same as the original MDP. Solving this BAMDP with the $\mathcal{T}^+$ yields the Bayes-optimal solution for the original MDP, balancing the exploration with exploitation optimally.

A multi-armed bandit problem can be expressed as a BAMDP where information states contain only the histories since there are no environment states. In that case, the history $h$ will consist of played arms and observed rewards so far. Let $h' = har$, where $h'$ is the played arm $a$ and observed reward $r$ appended to the history $h$. Then the transitions are defined as $\mathcal{T}^+(h, a, h') = \mathbb{E}_{P(\mathcal{R}|h)}[P(r|a)]$ where $P(\mathcal{R} \mid h)$ is the posterior belief about reward probabilities of all arms, given history $h$. The solution to this BAMDP, with a given prior $P(\mathcal{R})$ as the starting state, provides the Bayes-optimal solution to the multi-armed bandit problem.

One can derive popular MAB algorithms within the BAMDP as well. For instance, Thompson sampling procedure at each time step can be seen as estimating $\mathcal{T}^+$ with a single sample from $P(\mathcal{R} \mid h)$ instead of the full expectation, and then acting with a greedy argmax policy on it.

Different from the BAMDP formulations of multi-armed bandits, where the action space is choosing among the arms, the teaching MDP takes the perspective of the reward generating mechanism (teacher). While the uncertainty for a BAMDP agent is about the reward distribution, for a teacher, the uncertainty is about which arm the learner will sample next.

## 5   Algorithmic Overview for Bandit Learner

Algorithm 1 describes a bandit learner, with naive, planning, or mixture model of the teacher. The learner nests a model of the teacher (Algorithm 2). Note that the history $h_t$, up to $t$, is defined as the sequence $h_t = \boldsymbol{x}_1, y_1, \boldsymbol{x}_2, y_2, \ldots, \boldsymbol{x}_t$, which ends at the input $\boldsymbol{x}_t$ and doesn't include $y_t$. We also assume $p(\boldsymbol{\theta} \mid h_0, y_0) = p(\boldsymbol{\theta})$ in Algorithm 1. The naive, planning, and mixture likelihoods are defined in Equations 1, 4, and 5 in the main text.

**Algorithm 1** Bandit Learner
***
**for** $t \leftarrow 1$ to $T$ **do**
    $i_t \leftarrow$ thompson_sample$(p(\boldsymbol{\theta} \mid h_{t-1}, y_{t-1}), \boldsymbol{X})$          $\triangleright$ select the next arm
    $h_t \leftarrow h_{t-1}, y_{t-1}, \boldsymbol{x}_{i_t}$          $\triangleright$ update history
    $y_t \leftarrow$ teacher$(i_t)$          $\triangleright$ get the response from the teacher
    **if** learners_teacher_model = naive **then**
        $\mathcal{L} \leftarrow p_{\mathcal{B}}(y_t \mid \boldsymbol{x}_{i_t}, \boldsymbol{\theta})$          $\triangleright$ naive teacher likelihood
    **if** learners_teacher_model = planning **then**
        $\mathcal{L} \leftarrow$ planning_teacher_model$(h_t, y_t)$          $\triangleright$ planning teacher likelihood
    **if** learners_teacher_model = mixture **then**
        $\mathcal{L} \leftarrow (1-\alpha)p_{\mathcal{B}}(y_t \mid \boldsymbol{x}_{i_t}, \boldsymbol{\theta}) + \alpha$planning_teacher_model$(h_t, y_t)$    $\triangleright$ mixture likelihood
    $p(\boldsymbol{\theta}, \alpha \mid y_t, h_t) \leftarrow$ posterior_update$(\mathcal{L}, p(\boldsymbol{\theta}, \alpha \mid h_{t-1}, y_{t-1}))$    $\triangleright$ ($\alpha$ if mixture model)
***

**Algorithm 2** One-step Planning Teacher Model
***
**function** PLANNING_TEACHER_MODEL$(h, y)$
    **for all** $y' \in \{0, 1\}$ **do**
        $p_n(\boldsymbol{\theta} \mid h, y') \leftarrow$ naive_update$(h, y')$          $\triangleright$ simulate posterior update of naive learner
        $\boldsymbol{p}_{h,y'} \leftarrow$ estimate_thompson_probabilities$(p_n(\boldsymbol{\theta} \mid h, y'), \boldsymbol{X})$    $\triangleright$ next arm probabilities
    **return** $p_{\mathcal{M}}(y \mid h, \boldsymbol{\theta}) \propto \exp(\beta(\boldsymbol{\theta}^{\mathrm{T}} \boldsymbol{X}^{\mathrm{T}} \boldsymbol{p}_{h,y}))$          $\triangleright$ return the planning likelihood
***

# 6 Supplementary results for simulated experiments

## 6.1 Logistic regression active learner

While our main experiments focus on the multi-armed bandit setting, we provide here an additional experiment for teaching of a logistic regression active learner. This experiment uses the Wine Quality dataset [8], consisting of 4,898 instances of white wines with 11 continuous features and a ordinal output variable denoting wine quality. We transform the problem into a classification task by thresholding the quality.

The learner is a logistic regression model with L2 regularization. It is initialised with 2 data points (one for each label from the large clusters in the middle) and has a further pool of 2000 unlabeled data points for which it can query the label. The learner uses uncertainty sampling for selecting queries: the next query is chosen as the data point $\boldsymbol{x}$ for the label of which the current logistic regression model has the largest entropy, $-\sum_{y \in \{0,1\}} p_{\boldsymbol{\theta}}(y \mid x) \log p_{\boldsymbol{\theta}}(y \mid x)$. After obtaining a label, the model is updated and a new query done. Each unlabeled data point can be queried only once. The rest of the dataset is used as a test dataset for evaluating the performance, with classification accuracy as the performance metric.

The teacher plans for 1 step ahead, with full knowledge of the labels for the pool of 2000 data points (but not the test data). The reward is defined as the accuracy of the classifier.

We run the experiment for a horizon of 100 steps, with 100 repetitions of randomly dividing the data into the training pool and test set.

Figure 1 compares the accuracy on the test set for the active learning with teacher (green), active learner without teacher (orange), and learner using random queries to gather more data (blue). The accuracy of the logistic regression model fitted to the full pool is shown for reference (black). The teacher improves the learning performance markedly, attaining performance close to the full pool model with around 20 training samples.

## 6.2 Multi-armed bandits

We provide further results for the simulation studies here, as listed below. The two further datasets are the following, corresponding roughly to data that would occur in tasks for recommendation and image search, respectively. The Wine Quality dataset [8] consists of 4,898 instances of white wines with 11 continuous features (and the ordinal output variable denoting wine quality which is not used here).

Figure 1: Effect of teaching of pool-based logistic regression active learner in the Wine dataset. Lines show the mean over 100 replications and shaded area the 95% confidence intervals for the mean.

The Leaf dataset [9] consists of 340 instances with 14 features representing the shape and texture features of leaves from different plant species. For all datasets, all feature vectors are mean-centred and normalised to unit length.

Here, for the Word dataset, we also present a sensitivity analysis for a different ground-truth reward profile generated by $c = -2$ and $d = 6$, referred to as the supplementary setting. The qualitative difference between the new profile and the profile from the main text is that in the former there are more arms with high reward probabilities. This makes the rewards more informative and supportive of learning. The figures related to the supplementary setting indicate that a more informative reward means smaller gains from a planning teacher steering a naive learner. However, the combination of a planning teacher and a teacher-aware learner still provides a considerable improvement.

All simulation experiments were run on Linux computers, running PyTorch 1.0 and Pyro 0.3.

- Figure 2: The relevance profiles of the different datasets demonstrating the density of the reward probabilities. Word (Main) is the setting used in the results of the main text.
- Figure 3: Replication of the simulated experiment in the Word dataset using the supplementary setting.
- Figure 4: Replication of the simulated experiment in the Wine Quality dataset.
- Figure 5: Replication of the simulated experiment in the Leaf dataset.
- Figure 6: Concordance index results for the multi-step experiment from the main text.
- Figure 7: Replication of the simulated experiment in the Word dataset using the supplementary setting, with teacher's optimality parameter $\hat{\beta} = 5$ and teacher model parameter $\beta = 5$. This demonstrates that highly suboptimal teaching degrades learning performance.
- Figure 8: Replication of the simulated experiment in the Word dataset using the supplementary setting, with teacher's optimality parameter $\hat{\beta} = 10$ and teacher model parameter $\beta = 10$.
- Figure 9: Replication of the simulated experiment with 500 arms in the Word dataset using the supplementary setting.

Figure 2: The relevance profiles of the different datasets, generated by sorting the arms according to their reward probabilities and taking the mean over replications. For example, a point at (80,0.6) should be read as there are 20 arms with $\geq 0.6$ reward probability.

Figure 3: Replication of the simulated experiment for the supplementary relevance profile setting in the Word dataset.

Figure 4: Replication of the simulated experiment in the Wine Quality dataset.

Figure 5: Replication of the simulated experiment in the Leaf dataset.

Figure 6: Concordance index results for the multi-step experiment from the main text.

Figure 7: Replication of the simulated experiment for the supplementary setting in the Word dataset with teacher's optimality parameter $\beta = 5$.

Figure 8: Replication of the simulated experiment for the supplementary setting in the Word dataset with teacher's optimality parameter $\beta = 10$.

Figure 9: Replication of the simulated experiment for the supplementary setting with 500 arms in the Word dataset.

# 7 User study details

We conducted a proof-of-concept user study using a subset of 20 words (from the Word dataset) on ten university students and researchers (3 females and 7 males). The goal of the study was introduced to the participants as helping a system to find a target word, as fast as possible, by sequentially providing binary answers (yes/no) to the system's questions (15 question budget) about the relevance of different words to the target word. The target word was given to the participants at the beginning of each round. Since only 20 words were selected for the user study, we skipped the PCA pre-processing step (considering it is hard to detect information from noise when the number of data is much smaller than the dimension) and instead used a pairwise radial basis function kernel between the words in the original 300 dimension to reduce the dimension to 20. Furthermore, to reduce the variance in the sequence of questions in both models, we used Bayes-UCB [10] instead of Thompson sampling as the system's outer-most arm selection strategy. The list of considered words along with their feature vectors is shown in Figure 10. The resulting data matrix was also used as the ground truth reward function for each target word (asking about the target word gains reward one and others noticeably less than one).

The study was repeated for naive and mixture models (model means learner's model of the teacher) in randomised order and for twenty rounds (each word chosen once as the target word). The planning teacher model in the mixture had one-step planning horizon ($T = 1$). The starting question of each round was chosen randomly for each user and target but it was the same between the two models. Two practice rounds (one with each model) were completed in the beginning of the study. The user interface and time delays between questions were identical between the two models and the participants were naive about which system they were interacting with. The questions were in the form of "Is *word* Relevant?" and the participants could answer by typing "y" (Yes) or "n" (No) in the terminal and pressing enter. Each round would end after 15 questions and answers. The users were not under any time pressure and the study took on average 75 minutes.

The participants were compensated by a movie ticket upon completion of the study. The task performance was incentivised by providing an extra movie ticket if the participant was able to help the model find the target word in fewer steps than a certain threshold. All participants signed a standard consent form prior to the study. All user studies were performed on a Windows 10 laptop.

Figure 10: User study data and ground truth rewards. The matrix represents the feature vectors of each word considered in the user study. The ground truth reward values for each target word are represented by the values in the corresponding row.

Figure 11: P-value for paired sample t-test between average cumulative reward of the mixture and naive models of ten participants of the user study at each iteration. The black dashed lines show the 0.05 and 0.01 thresholds. The mixture model achieved significantly higher cumulative reward after 12 questions.