[Reviews · NeurIPS 2019]

Reviewer 1



This paper considers the problem of teaching an active sequential *machine learner* (e.g., an active learning algorithm), with a teacher which can “fake” the labels/outcomes of training examples with the goal of steering the learner faster to the goal state. The authors refer to such teacher a “planning teacher”, as opposed to a “naive teacher” which is often considered in the classical machine teaching problems. This setting differs from conventional machine teaching settings, in that In classical machine teaching setting, the teacher can only choose among a given set of training examples that are consistent with the target concept, and is often not allowed to provide inconsistent examples. The majority of the existing work in machine teaching considers teaching a “passive learner”, with a few exceptions (see additional reference in the comments below). The assumption that the teacher can choose the data-generation distribution makes it a very powerful teacher with a much richer action set than conventional teaching. This makes the problem considered in this paper an interesting scenario with useful applications. The formulation of the teaching problem as a planning problem over a teaching MDP is also interesting. However, the paper seems to lack strong theoretical support (e.g., to rigorously reason about the gain of the learner being aware of itself being taught), while the algorithmic contribution, including the experimental study, seems incremental. === Post-rebuttal comments The rebuttal to some extent strengthens the empirical results by including additional results on pool-based active learning. I'll be happy to revise my review to borderline score (4->5). I would also suggest the authors revise the title to reflect the specific settings (e.g., “planning teacher”) of the machine teaching problem. Overall I think the results could offer useful insights for practitioners working as it provides an interesting new direction in machine teaching. While I appreciate the existing empirical results demonstrating scenarios that a planning teacher would help, understanding why adding fake labels helps, and under what conditions it helps, seems an important question that remains undiscussed.

Reviewer 2



This paper frames the learning problem as a cooperative game between the machine and human, where the human has a mental model of the machine's task. In this situation, the learner poses queries (similar to query-driven active learning) and the teacher responds strategically to them, having knowledge the learner doesn't have, to best guide the learner. This is called "machine teaching". The learner may be aware of what the teacher is doing or not. Both cases are considered as well as a mixture model of the two. The learner then has an "inverse reinforcement learning" (IRL) problem to solve, to the teacher's - in this case the human user - "inverse machine learning" problem. The learner queries the teacher, who then makes a possibly strategic recommendation based on state knowledge that the learner does not have, then the learner makes a move knowing that the teacher has this knowledge. This model contributes to understanding of user models, where the user has a mental model of how the system engaged with should perform. In the case presented of a multi-armed bandit, the teacher's knowledge comprises the reward probabilities of individual arms, with model parameters \theta, which allows the teacher to strategically alter responses y_i, by means of an MDP. For this to work, one "bandit" must inform another, else were they independent then no strategy of offering the "wrong" y_i would help. Figure 2 offers an example of this; however how this may work in the general case begs more elaboration, beyond the claim that the teacher uses a modified reward based on R_t(h_t;\theta*). Exactly how does this work for Bernoulli bandits? Then should we assume that this modified reward, and specifically the \theta* is what the learner seeks via inverse reinforcement learning? What is complicated here is that the reward probabilities do depend on \theta (Equation 1), so shouldn't the learner be able to infer \theta without the actions of the teacher? Maybe another way to state this is that dependencies among bandits due to \theta should be evident in the naive case, irrespective of the teacher's strategy. An experiment revealing the differences in the sequence of y_i and R_i with for the naive versus strategic teacher would be revealing. Insights into the claimed success of this method are limited by the lack of a complete model that comprehends both teacher and learner in a cooperative game. Both apparently share the same "uber" reward to be optimized, perhaps with different horizons. As a cooperative game one might expect an equilibrium solution that reveals properties of their combined actions. Without a joint game theoretic model (perhaps by solving the I-POMDP) it's not apparent why the combination works. This is a fascinating area with rich implications for understanding human-machine behavior, but currently this paper is limited to computational results suggestive of general system properties. Specifics: The paper mentions available source code, but none was provided; perhaps an oversight?

Reviewer 3



Originality: The paper presents an MDP model of the machine teaching problem and builds a framework around this to improve learning with a teacher. Quality: Good quality work (idea, presentation, empirical studies). Clarity: Except for the abstract, I found the paper well written and easy to follow. Significance: A relevant contribution to machine teaching. Below I list some comments that provide further details on my review and might be helpful in improving the manuscript. Main comments: 1) From the abstract, I was not able to tell what the paper is about and what its main ideas and contributions are. Some examples of what did not become clear: - novelty and benefit of MDP model of teacher - key goal of using a model of the teacher to improve learning performance - point (1) is unclear I suggest rewording the abstract. 2) l. 98: Why is a deterministic learning algorithm considered? I suppose this is because otherwise, capturing the learner’s dynamics might be more challenging. But could one not also consider stochastic learning in the same framework? Maybe this could be an aspect to discuss. 3) Is the teaching MDP formulation of Sec. 3.2 a novel contribution (the review of related work seems to suggest this)? If yes, I suggest to explicitly state this somewhere in Sec. 3. 4) l. 139: Is the teacher’s reward the same as the reward previously defined for the learner? Specifically, is R_t(h_t) = R_t as in l. 110? Or does the teacher follow a different goal? This should be discussed and made explicit. If it's not the same, different letters should be used for the variables to avoid confusion. 5) l. 160: The choice of the reward function as the scalar product between next sample location and true parameters is unclear to me. Can you elaborate on why this choice? 6) Why is the model in (3) chosen? Specifically, the exponential terms with temperature parameter. Is this a common model in inverse RL? Minor: 7) ll. 49-50: It would be helpful if the authors added to this sentence, what exactly they do differently. How is another agent than the teacher involved in designing the learning data? 8) ll. 96-100: I find it slightly confusing that (1), (2), (3) are used for enumeration here, as parentheses usually denote equation references. Consider changing to, e.g., (i), (ii), ... Similarly, in other parts of the paper. 9) l. 101: T should be math font 10) Unless I missed this, "boundedly optimal teachers" is never explained in the paper. I suggest doing this. 11) ll. 200-206: This is a nice illustrative example and discussion. ============================================================ AFTER REBUTTAL: I have read the authors’ response. I appreciate the authors’ explanations, which address my questions and concerns in a satisfactory manner. I will thus keep my original score.

Reviewer 4



Advantages: This paper considers combining machine teaching with active learning. Under this setting, the active learner decides the instances to query, and the teacher decides the label to return. This paper discloses a non-trivial fact: the active oracle can be sub-optimal even if it generates the label through the underlying true distribution. This reveals the importance for combining active learning and teaching. To my understanding, this is the most significant contribution for the paper. Furthermore, the Bernoulli bandit (generalized linear bandit in fact) problem is taken as an example to show how effective teaching can be achieved. The MDP-based sequential decision framework is utilized to model the teaching-student learning process. On the high-level, the proposed approach seems to be a reasonable framework to solve this kind of problems. The most interesting part of the experimental results is that they show that if the student can model the teacher’s behavior properly, then the learning performance can be boosted. Furthermore, they also suggest the potential usefulness of the proposed framework under the human-involved oracle scenario. These two results could be very useful guidance for further studies. Disadvantages: Writing can be significantly improved. The main contributions are not clearly described, and the algorithmic parts are also not clear enough for understanding. The explanation of why providing labels through the underlying true distribution can be sub-optimal is insufficient. The bandit algorithm seems not correct. The description of how to estimate theta in line 114-119 is not clear. It seems that they rely on the assumption that the x_t received in all iterations are independent, while this is not true since the choice of arms are decided by the learner. The discussed bandit problem is actually a special case of the generalized linear bandit, I suggest referring to [1] for the Thompson sampling algorithm under this setting. Overall, the paper discusses an interesting and important problem. The novelty is significant, and it discloses several non-trivial facts. The high-level framework can be followed by future researches. While the writing can be significantly improved and the issue in the bandit learning algorithm should be addressed. This makes my score decreased (if I am wrong about the bandit algorithm, feel free to point it out) [1] https://arxiv.org/abs/1611.06534 ------ after rebuttal: I agree with the author response that the bandit algorithm in the paper follows from the general framework of TS. On the other hand, I find no theoretical guarantees of this algorithm under this setting both in the paper or in the literature. I made a mistake in the review before: the Bernoulli bandit in the paper is not the generalized linear bandit model since the randomness is not from the additive noise. In spite of the issues of lacking theoretical guarantees, I appreciate the novel insights from the paper, thus I would like to raise my score. I also encourage the authors to include more discussions about the possibilities of further theoretical studies.

[Author Response · NeurIPS 2019]

We would like to thank the reviewers for the constructive reviews.

**R1, R3**: *provide theoretical arguments.* **R6**: *"explanation why [...] true distribution can*
*be sub-optimal is insufficient."* – We agree that developing theory is important. Yet, the
focus of the paper is more conceptual and applied: we outline the new machine teaching
setting, propose solutions to it, and give empirical validation, including a user study.
We hope our paper will also entice others to contribute to this line of work, also more
theoretically. In ideal settings (exact computation, etc.), the solution to the teaching
MDP will (on average) be at least as good as the true distribution, up to the planning
horizon, for the objective determining the reward of the MDP (since by definition, the
solution maximizes that). The illustrative examples in the paper show cases where the
teacher can do better than the true data labels, and our empirical experiments show
improved performance.

**R1**: *empirical study in teaching an active learner* – Good point, our main focus was on
the bandit setting, but we will add the empirical results for teaching of the uncertainty-
sampling-based logistic regression active learner for the illustrative example (upper
figure on the right; full horizon planning teacher) as 4th panel to Fig 1, and for the Wine
dataset (lower figure on the right; 1-step planning teacher) to the supplement. Both use
independent test sets to measure performance, averaging over 100 repetitions of runs,
and the teachers optimize for prediction accuracy (based on having knowledge of the full pool, including labels).

**R1**: *"body of theoretical work on teaching a strategic (teacher-aware) learner [...] missing from the related work"*
– Thank you for the references, we will include them to the related works. Their theory has important assumptions such
as concept-consistent labels, which we relax. Thus, they are not directly applicable, albeit highly relevant.

**R3**: *"[...] the teacher's knowledge comprises the reward probabilities of individual arms, with model parameters*
$\theta$, *which allows the teacher to strategically alter responses* $y_i$, *by means of an MDP. For this to work, one "bandit"*
*must inform another [...] Exactly how does this work for Bernoulli bandits? [...]"* And question about inferring $\theta^*/\theta$.
– Indeed, we consider Bernoulli bandits with arm dependencies, where a response to an arm gives also information
about other arms' rewards. Knowing the arm dependencies allows the teacher to better predict which arms the learner is
more likely to query in the future, and thus direct the learner. The ground truth (reward generating) $\theta^*$ and $\theta$ are equal:
given enough queries, the learner can infer $\theta$ without a teacher. But a teacher can make the learning faster (and, further,
a teacher-aware learner can improve over baseline learner).

**R3**: *source code* – We will provide a link upon acceptance.

**R5**: *"1) From the abstract, I was not able to tell what the paper is about and what its main ideas and contributions*
*are. [...] 2) l. 98: Why is a deterministic learning algorithm considered? [...] "3) Is the teaching MDP formulation of*
*Sec. 3.2 a novel contribution [...] 4) l. 139: Is the teacher's reward the same as the reward previously defined for the*
*learner? [...] 5) l. 160: The choice of the reward function as the scalar product between next sample location and true*
*parameters is unclear to me. [...] 6) Why is the model in (3) chosen? [...]"* – Thanks, we will clarify and discuss these
in the revision: 1) We will rephrase the abstract to: first, clearly state the new machine teaching setting we propose;
second, our solutions to teaching problem and learning from a teacher; third, its advantages and increase in empirical
performance. 2) We focused on deterministic learning algorithms, but there is no inherent reason why stochastic
learning algorithms could not be considered, apart from making the problem more complex. 3) Yes; (PO)MDPs have
been used to model teaching, as we mention in the Related works, but not in a setting with an active sequential learner.
4&5) The teacher's reward is the same as the learner's expected reward (both include the scalar product for the chosen
arm $x_t$ at current iteration), except we didn't include the logistic function $\sigma(\cdot)$ in the teacher's reward to simplify the
formulas for the teaching model. We will clarify this in the revision. 6) The softmax policy is a common model in
probabilistic and MaxEnt reinforcement learning. It has also been shown to model human choice behaviour well.

**R6**: *clarity* – We will use the added 9th page for clearly commented pseudo-code algorithms.

**R6**: *"The bandit algorithm seems not correct. [...] assumption that the $x_t$ received in all iterations are independent"*
– It is correct; our Thompson sampling algorithm is the standard version used for Bayesian models (Alg. 4 in our ref. 34:
`https://arxiv.org/abs/1707.02038`), here sampling from the posterior of the generalized linear model (GLM).
We only assume the independence of the "noise" in the Bernoulli distributions generating the rewards at each time step,
thus the likelihood terms factor; on the other hand, the posterior distribution is conditioned on the observed arm feature
vectors $x_t$ and the distribution of $x_t$ is not modelled (as is usual for GLMs). We will add pseudo-code algorithms in the
revision, which will also make this clearer. Thanks for the GLM reference, we will add it in the revision.

**General**: – Thank you for the minor comments which we will naturally fix.

[Meta-Review · NeurIPS 2019]

The reviewers are mostly positive about accepting this work. The only significant reservation is that they would have liked to get a better intuitive understanding of why adding fake labels helps, and under what conditions it helps. However, they found the supplied empirical studies and a few toy examples to be sufficient evidence to support accepting the paper.